# Numerical and Experimental Study of the Fatigue Behavior for a Medical Rehabilitation Exoskeleton Device Using the Resonance Method

**DOI:** 10.3390/ma16031316

**Published:** 2023-02-03

**Authors:** Ana-Maria Mitu, Tudor Sireteanu, Nicolae Pop, Liviu Cristian Chis, Vicentiu Marius Maxim, Mirela Roxana Apsan

**Affiliations:** 1Institute of Solid Mechanics, Romanian Academy, 15 Constantin Mille Street, 010141 Bucharest, Romania; 2Engineering Faculty, Department of Industrial Engineering and Management, “George Emil Palade” University of Medicine, Pharmacy, Science, and Technology of Târgu Mureș, 38 Gheorghe Marinescu Street, 540139 Târgu Mureș, Romania; 3SC PROVETIM SRL, 313 Livezeni Street, 547365 Târgu Mureș, Romania; 4Vital SA, Water and Sewerage Utilities Company, 430311 Baia Mare, Romania; 5IKEA Purchasing Services SRL, 59 Grigore Alexandrescu Street, 010623 Bucharest, Romania

**Keywords:** dynamic modeling, modal analysis, computational mechanics, finite element method, random loads, fatigue, vibration fatigue, experimental techniques, degradation mechanisms, resonance method

## Abstract

In this paper, the dynamic behavior of a hip level joint device of an active exoskeleton used in the medical field is analyzed. The finite element method is used in order to determine the first frequencies and the eigenmodes, necessary for the fatigue testing in the resonance regime.

## 1. Introduction

The development of materials or structural components requires fatigue tests that more or less simulate operating conditions. Fatigue tests can be performed by two main methods: by applying a large number of cyclic loads or using the resonance method. Fatigue stress is performed by applying a large number of cyclic loads (forces or moments) on specialized stands to produce deformations (tensions) of the tested element, especially in the elastic domain. In general, these tests are performed on universal testing machines (usually servo-hydraulic), using standardized material samples and less structural components, due to assembly difficulties and dimensional constraints. Achieving the forces necessary to obtain the fatigue stress in this method involves energy consumption and test durations which, in general, are very large.

In this paper, the fatigue behavior of a metal component part of an exoskeleton is studied. The exoskeletons are used in the rehabilitation of different pathologies such as: stroke, diabetic foot, neuromuscular diseases and, last but not least, spinal cord injuries, as an active part of a patient’s process of rehabilitation as well as part of their future autonomy [1]. The study of the fatigue behavior of different parts of the exoskeleton is very important in order to improve their quality and lifespan.

In the present work, the fatigue stress is achieved by using the resonance phenomenon. The resonance regime is obtained by imposing cyclic displacements (linear or circular) in certain areas of the tested structure component, such as to excite some of its vibration modes, predictable by calculation. For most cases, the first natural mode is of particular interest, as the response of the structure to resonance in this mode has the greatest amplification [2]. 

The first natural frequency of the considered device is theoretically predicted by using the finite element method. The first approximation step consists in rewriting the elastic-dynamics equation with boundary and initial conditions from the classical approach to the variational form. This has an advantage, in that it contains the boundary conditions and the degree of the derivation operator is half reduced (Gauss’ formula and the space of admissible functions). The next step is the approximation of the spatial variable, from the variational form, using the finite element method, obtaining a second degree ordinary differential system with initial conditions. The last approximation stage consists in the approximation of the temporal variable, using the finite difference method, resulting in the nodal displacements, nodal velocities and nodal accelerations at each time step. After the first natural mode is achieved by numerical study, the experimental test is carried out by keeping the oscillations near the obtained resonance frequency. In this way, only a relatively small energy is required to maintain large response amplitudes, which can produce significant stresses in the tested structure. 

The load stress of the tested component can also be enhanced through the inertial effect of some light masses, mounted on the sample to be tested, with the effect of a certain decrease in the resonance frequency. They must be located in the vicinity of some antinodes of the excited vibration mode.

Due to the structural degradation of the tested sample, the resonance frequency decreases as the number of test cycles increases, resulting in a gradual decrease in the amplitude of the structure’s response and, implicitly, in the stresses to which it is subjected. By monitoring this phenomenon, the excitation frequency can be changed in a controlled manner to find a new resonance regime and readjust the response amplitude to its initial level. This procedure can be repeated as many times as necessary for the resonant frequency to fall below a certain imposed limit, until the failure of the tested device. This testing mode can be automatically controlled and can reduce the time required to cause the break by about 10 times, compared to the universal testing machine approach. 

The fatigue testing using the resonance method is not yet standardized [3], although it is currently applied in many scientific works [4,5,6,7,8,9,10,11,12,13,14,15,16,17,18,19,20] and there are some companies that produce specialized stands for specific application of resonance fatigue testing.

In this paper, we present a way of performing the fatigue tests and the results obtained by applying the method of resonance stressing of a junction piece at the level of the hip of an active exoskeleton used in the medical field.

This manuscript is organized as follows: Section 2 includes the elastic-dynamic equations in classical and in variational form, with boundary and initial conditions. There is a brief review of the finite element method and the numerical methods used to obtain the natural frequencies and natural modes; Section 3 contains a description of the fatigue tests and their results, in both graphic and numerical form, describes the experimental tests and shows the comparisons between numerical and experimental results. Section 4 suggests a possible standardization and future development of the resonance testing method. Section 5 draws the conclusion.

## 2. Materials and Methods

The purpose of this section is to calculate the first natural frequency of the device, necessary to determine the resonance zone, in order to apply the resonance method for the study of fatigue behavior. The part of the medical device analyzed in this paper is the junction piece at the hip level of an active exoskeleton [1], shown in Figure 1. This is the most stressed component of the exoskeleton.

The dynamic equilibrium equation for a continuous elastic medium will be approximated with the finite element method and with other numerical methods [21,22,23,24,25,26]. The finite element method is widely used in solving differential equations and has the following basic characteristics: the deformations inside the finite element can be uniquely approximated depending on the deformations of the nodes that define the finite element, by means of some interpolation functions, and the functions in the basis of the approximation space have much smaller bounded support, compared to the whole domain, resulting in a band matrix with non-zero elements grouped around the main diagonal, which involves a small calculation time.

In our case, we will use a finite element of thin plate-type (shell), and the discretized equation will be valid only until exit from resonance regime, because beyond the resonance zone, the stiffness matrix *K* changes, due to the material’s degradation.

### 2.1. Classical and Variational Formulation of the Problem of Elastic-Dynamics

Let us consider an elasticity body that at a given time, t = 0, occupies domain Ω⊂$R^{d}$, where d=2 or d=3. The boundary of the body is divided into two sub-regions such that $\partial \Omega \equiv \Gamma ={\bar{\Gamma }}_{U}\cup {\bar{\Gamma }}_{N}$, which are topologically open, and disjoint and $mes\left(\Gamma_{N}\right)>0$.

The displacement $\bar{u}\left(t,x\right)$ will be prescribed on $\Gamma_{U}$ and traction $\bar{h}\left(t,x\right)$ is to be given on $\Gamma_{N}.$ For the beginning, the boundary $\Gamma_{N}$ is considered without tensions. At the same time, the stress vector $\sigma^{\left(n\right)}(u)$ is defined, oriented outwards of the boundary ∂Ω≡Γ, n is the outward normal unit vector on Γ.

The initial displacement $u\left(0,x\right)=u_{0}\left(x\right)$, the initial velocity $\dot{u}\left(0,x\right)=u_{1}\left(x\right)$ and the density of the volume force *f* are also given. The field of the displacements will be the solution, which must be found, of the differential equations of elastic-dynamics. The elastic-dynamics equation on Ω in a time interval $\left[0,t_{E}\right]$ with $t_{E}>0$, has the following form:(1)$$\rho \ddot{u}\left(t,x\right)-\sigma_{ij,j}\left(u\left(t,x\right)\right)=f\left(t,x\right)\;\mathrm{on}\;\left[0,t_{E}\right]\times \Omega .$$

The boundary conditions
(2)$$u\left(t,x\right)=\bar{u}\left(t,x\right)\;\mathrm{on}\;\left[0,t_{E}\right]\times \Gamma_{U},$$
(3)$$\sigma^{\left(n\right)}\left(u\right)\left(t,x\right)=\bar{h}\left(t,x\right),\;\mathrm{on}\;\left[0,t_{E}\right]\times \Gamma_{N},$$
where ρ is mass density with $\frac{\partial \rho }{\partial t}=0,\;\rho \in L^{\infty }(\Omega ),\;\rho \geq \rho_{0}>0$, $\ddot{u}\equiv \frac{\partial^{2}u}{\partial t^{2}}$ is acceleration.

The strains are given by relations $\varepsilon_{ij}(u)=\frac{1}{2}\;\left(\frac{\partial u_{i}}{\partial x_{j}}+\frac{\partial u_{j}}{\partial x_{i}}\right)$.

We shall use the following notations for the normal and tangential components of the displacements and of the stress vector:

$u_{N}=u\cdot n=u_{i}n_{i}\;$, $u_{T}=u-$$u_{N}n$, $\sigma_{N}=\sigma_{ij}n_{i}n_{j}$, ${\left(\sigma_{T}\right)}_{i}=\sigma_{ij}-\sigma_{N}n_{i}$, 1≤i,j≤d and n is the outward normal unit vector on Γ.

Following the steps, similar with those of [21], the linear elastic-dynamics problem can be formally equivalent with a variational problem. This equation has the advantage of containing the boundary conditions, thanks to the Gauss formula applied on the space of admissible functions, and the degree of the derivation operator is half reduced:

**Problem 1.** *Find the function* $u:[0,t_{E}]\to V$*so that*(4)$$\langle \ddot{u}\left(t\right),v-\dot{u}\left(t\right)\rangle +a\left(u\left(t\right),v-\dot{u}\left(t\right)\right)=\langle f\left(t\right),v-\dot{u}\left(t\right)\rangle ,\;\forall v\in V,$$*with the initial conditions:*(5)$$u\left(0,x\right)=u_{0}\left(x\right)\mathrm{and}\;\dot{u}\left(0,x\right)=u_{1}\left(x\right).$$

It is assumed here, for simplicity, that ρ≡1. The following notations and definitions were also used:(6)$$V=\left\{v\in {\left[H^{1}\left(\Omega \right)\right]}^{d}|\;v=\bar{u}\;a.e.\;on\;\Gamma_{U}\right\}$$ The space of admissible displacements (velocities);

a:V×V→ℝ, $a\left(u,v\right)=\int_{\Omega }C_{ijkl}\varepsilon_{ij}(u)\varepsilon_{kl}(v)dx$, the virtual work produced by the action of the stress $\sigma_{ij}\left(u\right)$ on the strains $\varepsilon_{ij}\left(v\right)$;

$f\left(t,x\right)\in V^{\prime },\;\langle L,v\rangle \equiv \langle f\left(t,x\right),v\rangle =\int_{\Omega }f\left(t,x\right)vdx+\int_{\Gamma_{N}}\bar{h}(t,x)\gamma_{T}\left(v\right)ds$ the virtual work produced by the external forces.

Here, 〈⋅,⋅〉 denotes duality pairing on $V\times V^{\prime }$ where $V^{\prime }$ is the topological dual of V. γ is the trace operator mapping from ${\left[H^{1}\left(\Omega \right)\right]}^{d}$ onto ${\left[H^{\frac{1}{2}}\left(\Gamma \right)\right]}^{d}$ which may be decomposed into a normal component $\gamma_{N}\left(v\right)$ and tangential component $\gamma_{T}\left(v\right)$.

### 2.2. Finite Element Approximations of the Elastic-Dynamics Problem

Two types of semi-discrete approximation scheme can be applied. In the first type, when we replace the infinite-dimensional space V by a finite-dimensional subspace $V_{h}$ (the element finite spaces and *h* characterize the size of the partition with the finite element), leading to a finite-dimensional system of ordinary Equation (9). In the second type, we replace the time derivatives by finite differences that lead to elliptic variational equations over infinite-dimensional space V at each time step. Such approximation schemes can be termed as temporally semi-discrete schemes. The error induced by finite element approximation can be minimized, by diminishing the size of the partition, *h*, or by increasing the degree of the element shape functions.

Using standard finite element procedures, an approximate version of Problem P1 can be constructed in finite-dimensional subspaces $V_{h}\left(\subset V\subset V^{\prime }\right)$. For certain (h) the approximate displacements, velocities and accelerations at each time t are elements of $V_{h}$, $v^{h}\left(t\right),{\dot{v}}^{h}\left(t\right),{\ddot{v}}^{h}\left(t\right)\in V_{h}$.

Within each element $\Omega_{h}^{e}\;\left(e=1,\ldots ,N_{h}\right)$, *N_h_* being the total number of finite elements, the components of the displacements, velocities and accelerations are expressed in the form:(7)$$v_{k}^{h}\left(t,x\right)=\sum_{I}^{N_{e}}v_{k}^{I}\left(t\right)N_{I}\left(x\right),\;{\dot{v}}_{k}^{h}\left(t,x\right)=\sum_{I}^{N_{e}}{\dot{v}}_{k}^{I}\left(t\right)N_{I}\left(x\right),\;{\ddot{v}}_{k}^{h}\left(t,x\right)=\sum_{I}^{N_{e}}{\ddot{v}}_{k}^{I}\left(t\right)N_{I}\left(x\right),$$
where $k=2\;or\;3,\;N_{e}=$ the number of the nodes of the element, $v_{k}^{I}\left(t\right)$, ${\dot{v}}_{k}^{I}\left(t\right)$, ${\ddot{v}}_{k}^{I}\left(t\right)$ are the nodal values of the displacements, velocities and accelerations, respectively, at the time t and $N_{I}$ is the element shape function associated with the nodal point I.

The finite element version of the Problem 1 is then:

**Problem 2.** *Find the functions* $u_{}^{h}:\;\left[0,t_{E}\right]\to V_{h}$*so that*(8)$$\langle {\ddot{u}}_{\varepsilon }^{h}\left(t\right),v^{h}\rangle +a\left(u_{\varepsilon }^{h}\left(t\right),v^{h}\right)=\langle f\left(t\right),v^{h}\rangle ,\;\forall v^{h}\in V_{h},$$*with the initial conditions (5)*.

If $N_{h}^{\Omega }$ is the number of the nodes of finite element mesh of Ω, then this problem is equivalent to the following matrix problem:

**Problem 3.** *Find the function* r:[0,tE]→ℝd×NhΩ, *so that*(9)M r¨(t)+Kr(t)=F(t),*with the initial conditions*(10)r(0)=r0, r˙(0)=r1.

Here, we have introduced the following matrix notations:

$r\left(t\right),\;\dot{r}\left(t\right),\;\ddot{r}\left(t\right)$: the column vectors of nodal displacements, velocities and accelerations, respectively;

M: mass matrix; 

K: stiffness matrix;

F(t): consistent nodal exterior forces vector;

The stiffness matrix K is obtained by assembling the stiffness matrices on each finite element, i.e.:$$K=\sum_{e=1}^{N_{h}}K_{e},$$
where $K_{e}$ is the stiffness matrix of the finite element e, and assembly will be done by identifying the global and local degrees of freedom of the finite element which will be assembled.

The vector of external forces, or the vector of generalized loads, is calculated similarly:$$F(t)=\sum_{e=1}^{N_{h}}F_{e}(t),$$
where $F_{e}(t)$ is the vector of external force of the finite element e from time t, and the assembly rule is similar to that of the stiffness matrix.

The mass matrix M will be a diagonal matrix, because the lumped mass technique is used, where the assembled mass matrix is the sum of the mass matrices of the finite elements, to which the concentrated masses specified in the degrees of freedom of the structure must be added. We remind you that for each finite element, the density of the material and the geometry are known, therefore, the volume and mass of each finished element can be calculated.

The thin plate element is, geometrically, a convex quadrilateral consisting of four compatible triangles. These triangles have bases on the sides of the quadrilateral, and the common vertex is designated by the intersection of the two segments that join the means of the opposite sides, called the central node. The central node will add to the quadrilateral element six internal degrees of freedom that will be eliminated by the static condensation process, therefore, the resulting quadrilateral element will have 24 degrees of freedom, (i.e., three displacements and three rotations, per node). The thickness of the thin plate-type finite element will be specified to each finite element, specifying that the four nodes that define the element are considered to be in the middle of the plate thickness.

Each thin plate-type quadrilateral finite element is defined by four nodal points I, J, K, L in a counter-clockwise direction, which form a convex quadrilateral. (Ox, Oy, Oz) is the element’s local system, defined as follows: O is the algebraic mean of the four nodes, I, J, K, L and the direction of Ox is specified by LI-JK, where LI and JK are the midpoints of the sides L-I and J-K; Oz is perpendicular to X-axis, and to the line joining the midpoints IJ and KL, at the point O; Oy is perpendicular to Ox and Oz, in the point O to complete the triorthogonal system, as shown in Figure 2.

### 2.3. Determinant, or Characteristic Polynomial, Method

First, we will solve the generalized problem of eigenvalues which results from free and undamped motion; thus, Equation (9) becomes:(11)$$M\;\ddot{r}\left(t\right)+Kr\left(t\right)=0.$$

For Equation (11), solutions of the form are sought:{r(t)}={X}cos(ωt+ψ), and it is necessary to check Equation (11). Thus, a linear and homogeneous algebraic system is obtained,
(12)$$(K-\omega^{{}^{2}}M)\left\{X\right\}=0.$$

It is known that such a system admits a non-zero solution, if and only if:(13)$$\det(K-\omega^{{}^{2}}M)=0.$$

Expanding the determinant in Equation (13) according to the powers of $\omega^{{}^{2}}$, a polynomial of order *n* is obtained, assuming that *n* is the size of the system. The eigenvalues will be the roots ω of the characteristic polynomial:(14)$$p(\omega^{2})=\det(K-\omega^{2}M).$$

Moreover, the Sturm separation theorem is used, checking the increasing sequence of all eigenvalues, $\omega_{i}$, i=1,⋯,n. The eigenvectors corresponding to the eigenvalues (eigenmodes) are calculated, by solving a homogeneous linear algebraic system, corresponding to each eigenvalue $\omega_{i}$. The components of the vectors $\;\left\{X_{i}\right\}$ will not be linearly independent, and for this reason, the magnitudes of the vectors cannot be obtained, but only their direction. By scaling, the modal matrix is obtained.

It is noted: Φ the matrix of M-orthogonalized eigenvectors with *p* = n, or *p* < n
(15)$$\Phi =[\Phi_{1},\Phi_{2},\cdots ,\Phi_{p}]$$
and Ω is the diagonal matrix with the squares of the eigenvalues (eigenfrequencies).
(16)$$\Omega^{2}=diag(\omega_{i}^{2}),$$
where $\omega_{i}^{2}$ are the squares of eigenvalues corresponding to the eigenvectors (eigenmodes).

Finally, a series of pairs of values and eigenvectors is obtained, starting with the dominant pair, in increasing order of eigenfrequencies, $\left(\omega_{1},\Phi_{1}),\;(\omega_{2},\Phi_{2}),\cdots ,(\omega_{2},\Phi_{p}\right)$.

In the analysis of the dynamic response modeled by Equation (9), F(t), can be a vector of time-varying loads or, in our case, loads resulting from the oscillatory movement of the hydropulse on which the test device is fixed. Assuming that the test device is uniformly subjected to the hydropulse acceleration, denoted by ${\ddot{r}}_{h}(t)$, the equilibrium Equation (9), becomes, given that F(t) becomes $-M\;{\ddot{r}}_{h}(t)$:(17)$$M\;{\ddot{r}}_{r}\left(t\right)+Kr_{r}\left(t\right)=-M\;{\ddot{r}}_{h}(t),$$
where $r_{r}\left(t\right)$ is the relative displacement of the device with respect to the hydropulse, i.e., $r_{r}\left(t\right)=r(t)-r_{h}(t)$.

### 2.4. Dynamic Response with the Modal Superposition Method

In the case of the modal superposition method, it is assumed that the device can be correctly modeled by the first p lowest vibration modes, where p << n. Using the transformation r(t)=ΦX, (see [26]), where the matrix Φ contains the first p orthonormalized M-eigenmodes are contained, $\Phi_{i}^{T}M\Phi_{j}=\delta_{i}^{j}$, Equation (9) becomes:(18)$$\ddot{X}+\Omega^{2}X=\Phi^{T}F(t).$$

Equation (18) represents a system of decoupled second-order ordinary differential equations. This can be solved relatively easily, using Wilson θ-method, which is an unconditionally stable step-by-step integration method.

For our case, with the prescribed hydropulse movement, $r_{r}(t)=\Phi X$, and Equation (18) will have the term on the right-hand side given by $-\Phi^{T}M\;{\ddot{r}}_{h}(t)$. It is known that the acceleration of the hydropulse is considered as the sum of the components on x, y and z, as prescribed.

## 3. Results

The set of equipment used for experimental study in the Dynamic Testing Laboratory from the Institute of Solid Mechanics is presented in Figure 3.

The resonance fatigue tests were carried out using a hydropulse Schenck PSA 100KN to excite the bending vibrations of tested junction piece, clamped rigidly on the hydropulse piston as a cantilever beam. The embedding system is shown in detail in Figure 4. This assembly allows us to obtain a significant cyclic bending stress of the tested part in resonance mode, by controlling the frequency and amplitude of the piston displacement.

The accelerometers placed on the hydropulse piston and on the exoskeleton component allow the evaluation of the amplification of the dynamic response of the tested structure, in order to realize its excitation in the resonance regime. 

The main characteristics of the PZT accelerometer, placed on the tested component, used for measurement of output vibration parameters, are summarized in Table 1.

The output signals of the PZT accelerometers are routed to charge conditioning amplifiers, having the following main features:Three digit conditioning to transducer sensitivity;Unified output ratings for simplified system calibration;High sensitivity up to 10 V/pC;Built-in integrators for displacement and velocity;Switchable low and high frequency limits.

The output voltage signals of charge amplifiers are supplied to a dual channel PC oscilloscope for analog–digital conversion. The digital data are further processed to obtain the measured vibration parameters in both time and frequency domains. 

The voltage output of the signal generator (controlled by the same PC oscilloscope) is routed to the hydropulse control unit.

### 3.1. Experimental Tests

The method of carrying out the fatigue tests by the resonance method, used in this paper, consisted of:predicting by FEM method the frequency of the first vibration mode for initial configuration of experimental setup;monitoring the evolution of the amplitude or the effective value of the response, by changing the excitation frequency in the neighborhood of predicted resonance frequency;determining the frequency spectrum of the response in the resonance regime for the evaluation of the resonance frequency;maintaining the frequency of the stress cycles at the value established in the previous step until the moment when the response vibration level of the tested component reach a prescribed percentage of the initial value due to the structural degradation of the material through fatigue;determining the frequency response function of the component by sweeping the excitation frequency, to obtain the value of the new resonance frequency;continuing the tests according to the previous steps until the failure (breakage) of tested piece.

The number of stress cycles is determined based on test durations and excitation frequencies, kept constant between two successive resonance values.

The tests were carried out successively in two stages:In the first stage, the tests were carried out with a small additional mass (acceleration transducer and mounting magnet with a total weight of 0.060 kg), placed on the sample, as shown in Figure 4. After a relatively large number of cycles at the initial resonance frequency (approx. 195 Hz), predicted by calculation with MEF, no significant decrease was observed.In the second stage, the tests carried out for the same sample on which a cylindrical body was mounted. The total additional mass was 0.620 kg (close to the mass of the exoskeleton drive motor, which is mounted on the junction piece, as shown in Figure 5). This testing setup simulates the operating conditions more realistically. The initial resonance frequency decreased to approx. 120 Hz, as was predicted by calculation with MEF and obtained experimentally. Fatigue testing started with the new resonance frequency, and was continued using the methodology previously described, until the failure (breaking) of the exoskeleton junction piece.

### 3.2. Results of the Fatigue Tests Performed by the Resonance Method

In this paragraph, the amplitude spectra and frequency response functions, recorded for different resonance regimes of tested sample, are presented. These records highlight the evolution of structural degradation versus number of stress cycles applied until the breaking of the sample in the clamped zone. It is worth mentioning that in the same zone were encountered breakings of exoskeleton joint pieces in operating conditions. 

In Figure 6, Figure 7 and Figure 8 are given the frequency response functions (amplification factors) of the junction component, tested with the additional mass of 0.060 kg. These plots are displayed on the PC oscilloscope screen, being obtained by automatic frequency sweep of the imposed stress cycles. Their values are expressed as the root mean square voltage (Vrms), supplied by the signal conditioners, being proportional to the measured vibration level.

As one can see, after a relatively large number of cycles at the resonance frequencies (approx. 1,370,000), only 0.01% decrease of initial value *f*_0_ = 195.3 Hz was observed. 

In order to accelerate the structural degradation, the tests were continued with an additional mass of 0.620 kg. This weight is close to that of the exoskeleton drive motor mounted on the tested component in operating conditions (see Figure 5).

In Figure 9, Figure 10, Figure 11, Figure 12, Figure 13, Figure 14, Figure 15, Figure 16, Figure 17, Figure 18 and Figure 19 are plotted the experimental amplitude spectra obtained for the measured vibrations of device with additional mass of 0.620 kg, recorded after different moments of the fatigue test. In the second stage of resonance fatigue testing, the evolution of structural degradation was more rapid than in the first stage and the results could have been affected by the relatively long time required for recording the frequency response functions by frequency sweeping. The amplitude spectra are almost instantaneously displayed after the end of recording the vibration time histories. The results are calculated in decibels voltage values (dBV) relative to the reference specified in the upper left side of plots. The sequence of the plots recorded in the second stage illustrates better the fatigue resonance methodology than those recorded in the first stage, due to the higher gradient of structural degradation. 

The frequency of the stress cycles was maintained at the value of 63.5 Hz for just 47,880 cycles until the vibration amplitude of the tested sample decreased very fast producing the breaking of it. Just before the end of the testing second stage, the last frequency response function was recorded at which the resonance regime could be obtained before breaking.

The above experimental results show a gradual decrease of the first vibration mode frequency of the tested device. After 1,062,126 cycles with the additional mass of 0.620 kg, the fatigue tests caused the failure of the exoskeleton junction piece. 

The variation of the resonance frequency in relation to the number of cumulative stress cycles for the two testing stages, mentioned in the previous paragraph, is presented in Figure 18. Practically, as one can see from this figure, the tested device failed in stage 2 in resonance mode at 114.8 Hz, after approx. 735,400 cycles.

The decrease of the resonant frequency of the device with the increase of the number of fatigue cycles could be viewed as a very sensitive measure of the material stiffness degradation. This stiffness degradation is frequently employed as a macroscopic measure of the fatigue damage degradation of the material in terms of macroscopic variables [15]. In the one-dimensional case (considering the device as a cantilever beam), the macroscopic damage variable $D_{N}$ is defined as
(19)$$D_{N}=1-\frac{E_{N}}{E_{0}}=1-{\left(\frac{f_{N}}{f_{0}}\right)}^{2}$$
where $E_{0}\;$ is the longitudinal stiffness of the undamaged device, $E_{N}$ is the longitudinal stiffness of the device after N fatigue cycles, $f_{0}$ is first eigenfrequency of the undamaged device and $f_{N}$ is first eigenfrequency of the device after N fatigue cycles. The variation of the damage variable $D_{N}$ versus the number of cycles N, determined from the diagram shown in Figure 18, is presented in Figure 19.

Figure 20 captures the breaking moment of the tested device.

Figure 21 and Figure 22 show the components of the broken sample and the appearance of the breaking surfaces.

## 4. Discussion

Due to the benefits of fatigue resonance testing, which is very useful in many applications, it is becoming necessary to standardize this method. 

The real-time monitoring of the evolution of resonance frequency and number of cycles by feedback control of the input and output parameters maintain the desired level of stress cycles.

We think the fatigue resonance testing could be efficiently applied to studying the effects of overload on fatigue life of notched specimens (e.g., the exoskeleton junction piece studied in this paper).

Investigation of the influence of additional mass placed on tested structural components could provide useful information for designing and resonance fatigue testing. In our case, reducing the weight of the mass of the exoskeleton drive motor could extend the life span of the exoskeleton junction piece.

## 5. Conclusions

The tests presented in this paper proved the effectiveness of the resonance method for testing the fatigue behavior of structural components.The first natural frequency of the sample, which is used initially as an imposed fatigue stress frequency for a given testing setup, was theoretically predicted using the finite element method and experimentally validated.The decrease of the resonant frequency of the device with the increase of the number of fatigue cycles is a measure of the material stiffness degradation.The fatigue tests, performed successively on the same specimen with two different additional masses (0.06 kg and 0.620 kg), highlighted a significant difference in the evolution of the macroscopic damage variable: 1% for approx. 1,380,000 cycles in the first case and 84% for approx. 1,060,000 cycles (until the breaking of the tested specimen).The appearance of the fracture surfaces highlights the ductile fracture of the sample and the way the cracks propagate from the areas of the stress concentrators near the connections to the line of the two holes applied in the vicinity of the clamped area, as predicted by FEM calculation.

## Figures and Tables

**Figure 1 materials-16-01316-f001:**
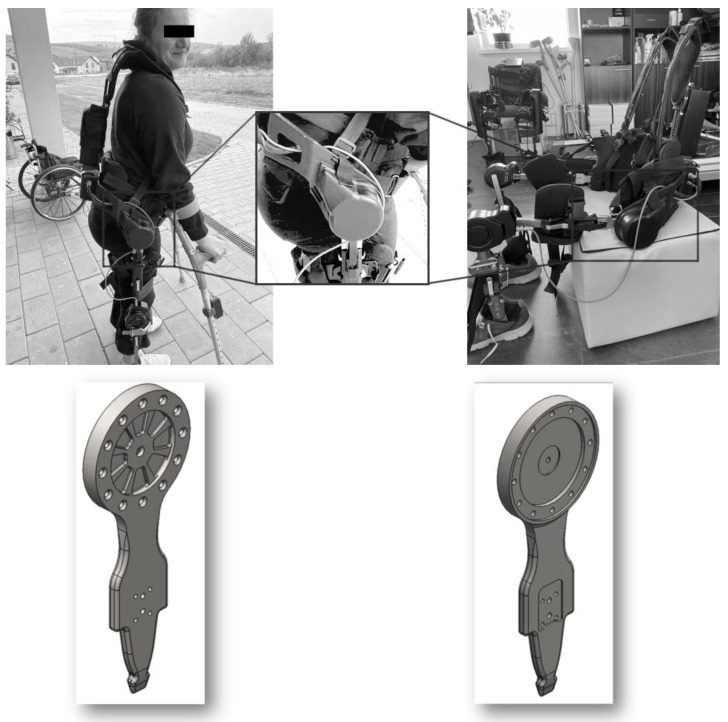
The exoskeleton and the tested junction piece.

**Figure 2 materials-16-01316-f002:**
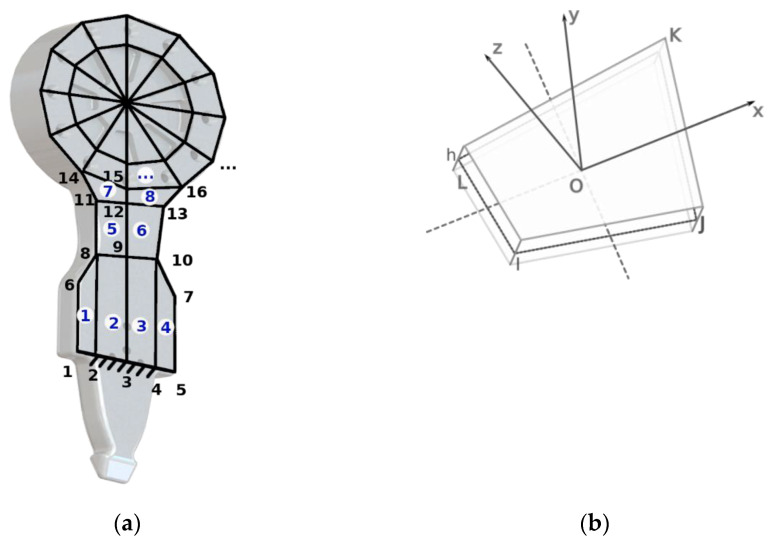
(**a**) The finite element mesh for the device; (**b**) Thin plate-type quadrilateral finite element used in discretization.

**Figure 3 materials-16-01316-f003:**
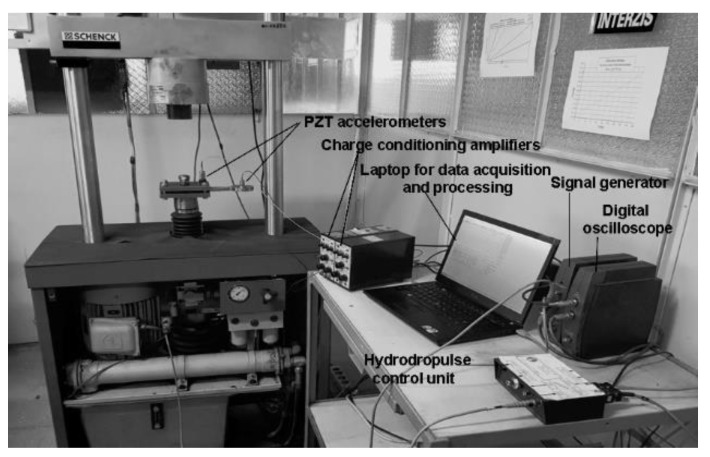
Testing rig and equipment for measurement, acquisition and processing of experimental data.

**Figure 4 materials-16-01316-f004:**
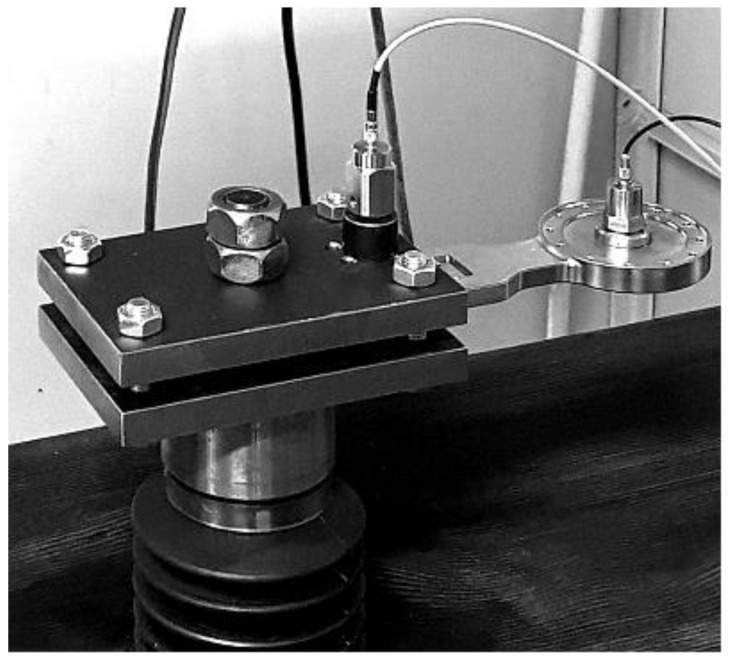
The device mounted on the hydropulse.

**Figure 5 materials-16-01316-f005:**
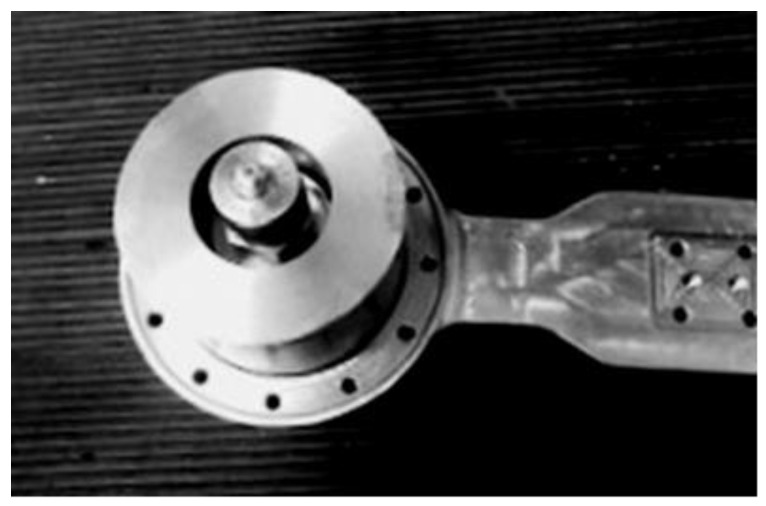
The sample with additional mass of 0.620 kg.

**Figure 6 materials-16-01316-f006:**
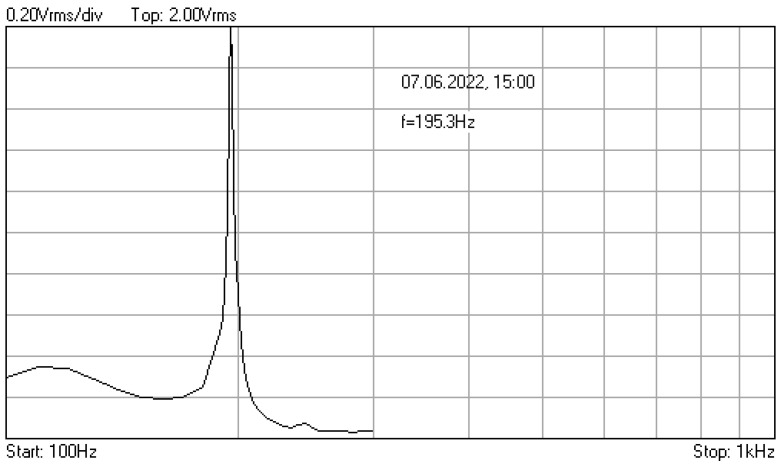
Frequency response function for additional mass 0.06 kg, recorded at the initial moment.

**Figure 7 materials-16-01316-f007:**
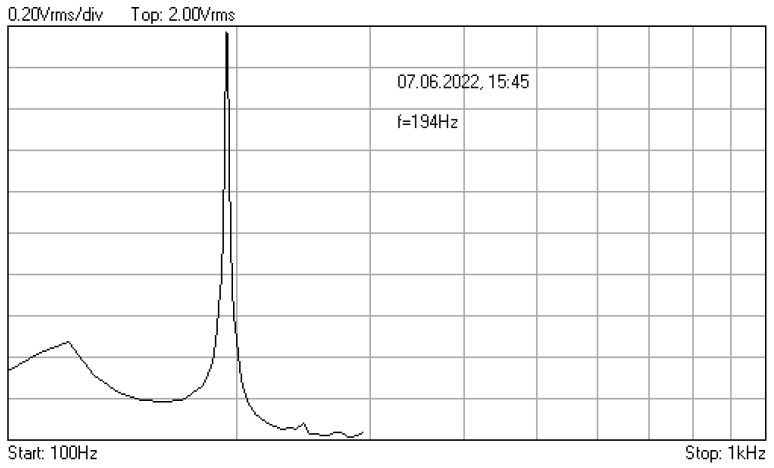
Frequency response function recorded after 843,696 cycles with frequency 195.3 Hz.

**Figure 8 materials-16-01316-f008:**
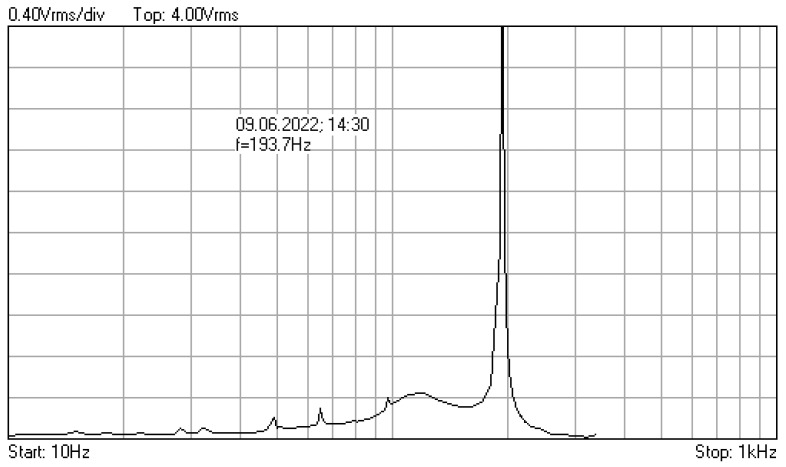
Frequency response function recorded after 523,800 cycles with frequency 194 Hz.

**Figure 9 materials-16-01316-f009:**
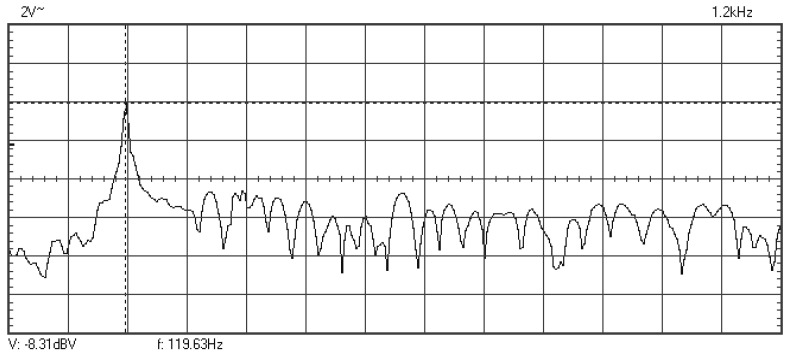
Amplitude spectrum for additional mass of 0.620 kg, recorded at the initial moment.

**Figure 10 materials-16-01316-f010:**
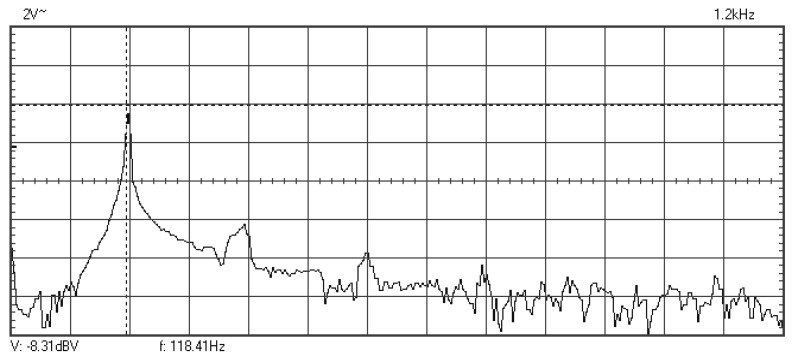
Amplitude spectrum recorded after 358,800 cycles at 119.6 Hz.

**Figure 11 materials-16-01316-f011:**
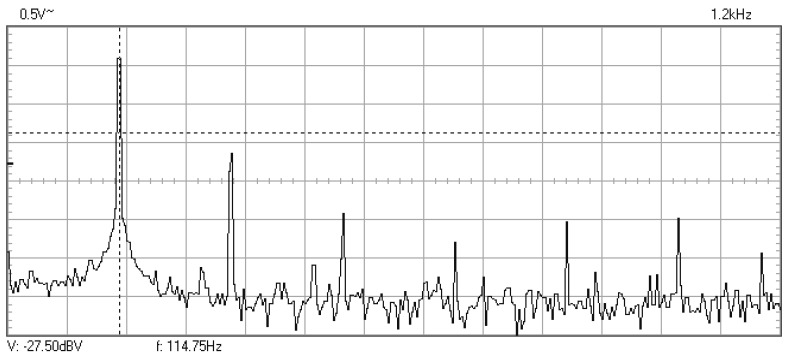
Amplitude spectrum recorded after 376,512 cycles at 118.4 Hz.

**Figure 12 materials-16-01316-f012:**
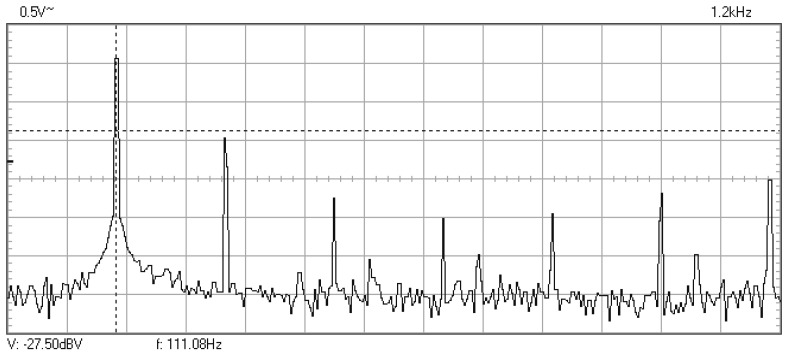
Amplitude spectrum recorded after 61,992 cycles at 114.8 Hz.

**Figure 13 materials-16-01316-f013:**
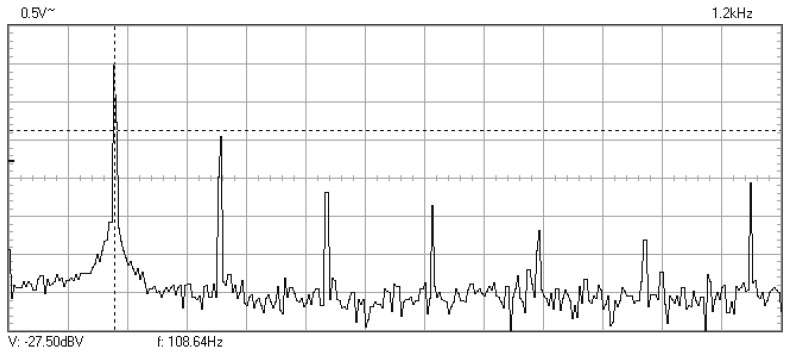
Amplitude spectrum recorded after 13,332 cycles at 111.1 Hz.

**Figure 14 materials-16-01316-f014:**
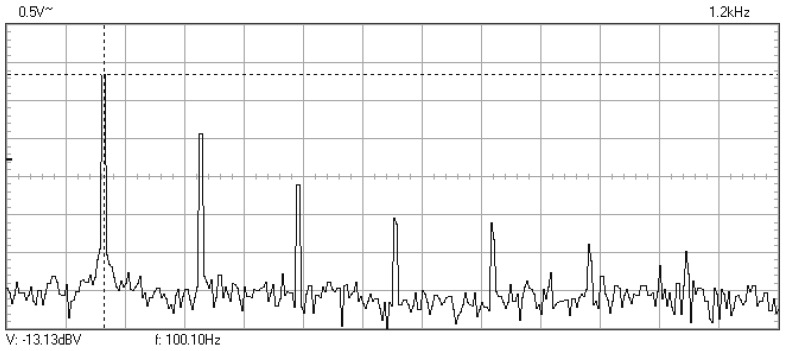
Amplitude spectrum recorded after 32,580 cycles at 108.6 Hz.

**Figure 15 materials-16-01316-f015:**
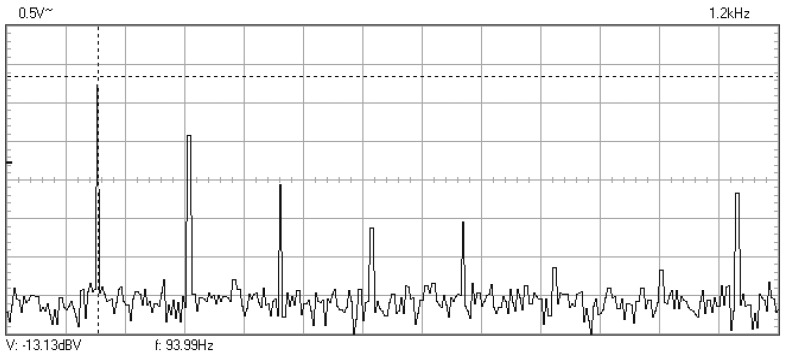
Amplitude spectrum recorded after 30,030 cycles at 100.1 Hz.

**Figure 16 materials-16-01316-f016:**
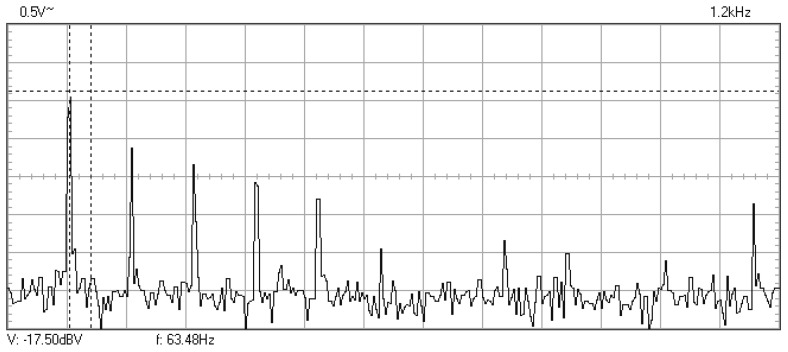
Frequency spectrum with the mass of 0.620 kg recorded after 141,000 cycles at 94 Hz.

**Figure 17 materials-16-01316-f017:**
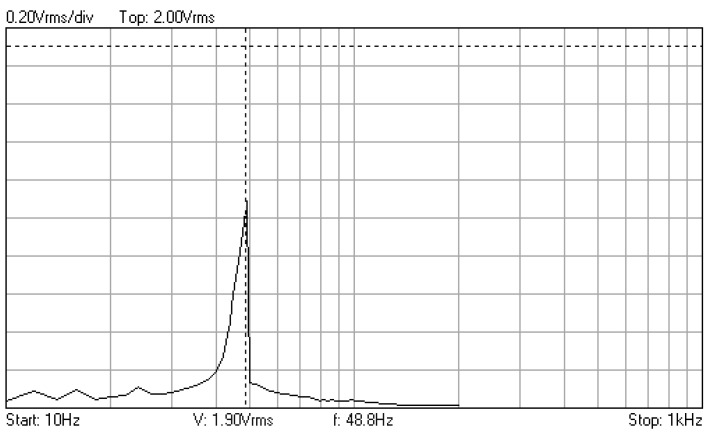
The last frequency at which the resonance regime could be obtained before breaking.

**Figure 18 materials-16-01316-f018:**
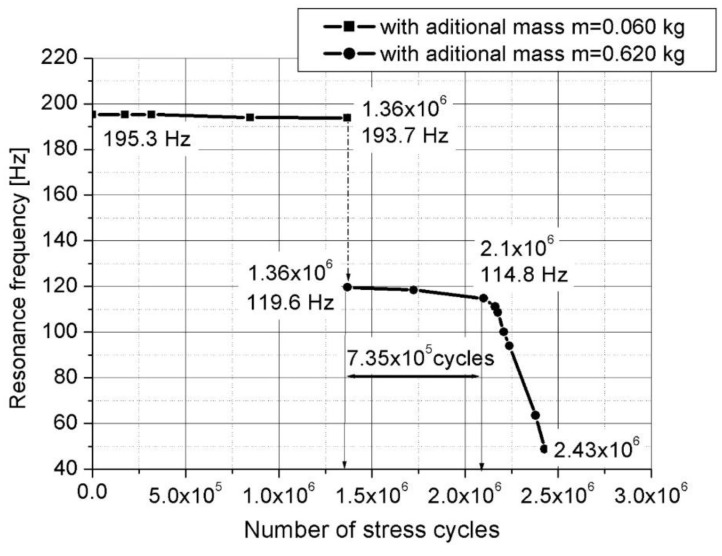
The variation of the resonance frequency versus the number of cumulative cycles.

**Figure 19 materials-16-01316-f019:**
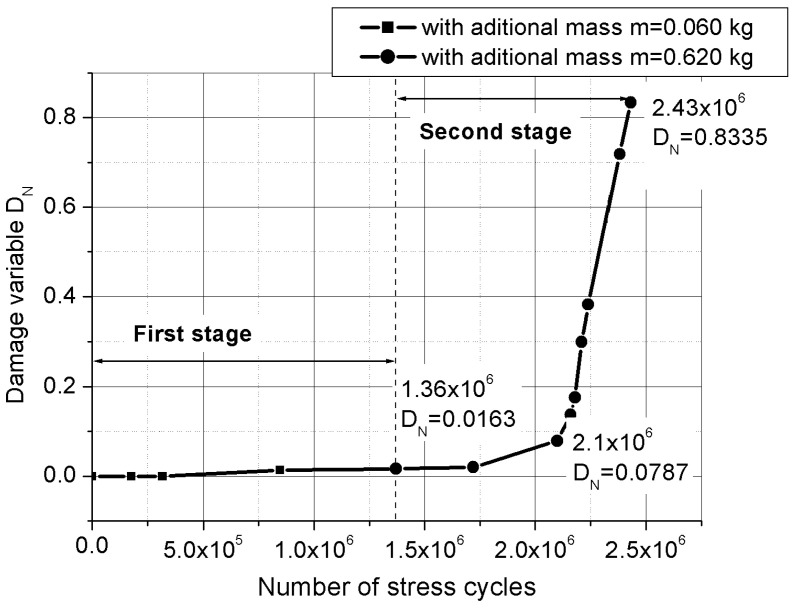
Evolution of the macroscopic damage variable with number of cumulative cycles.

**Figure 20 materials-16-01316-f020:**
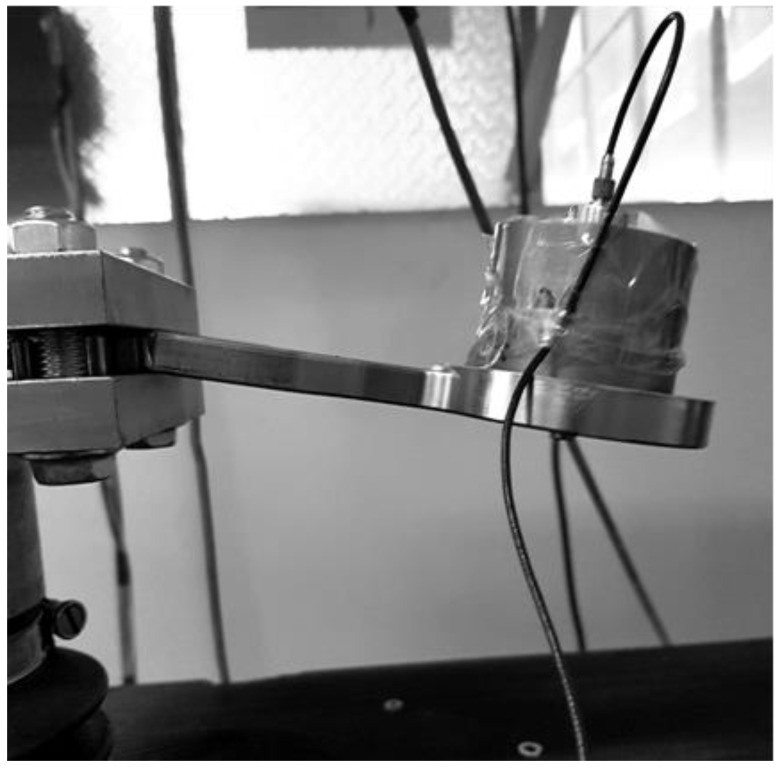
The tested device with 0.620 kg additional mass at breaking moment.

**Figure 21 materials-16-01316-f021:**
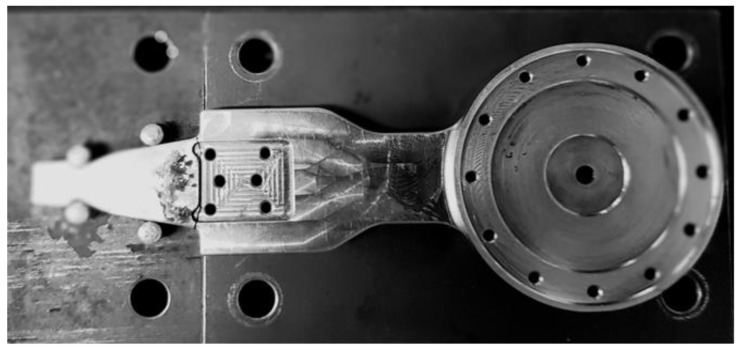
The breaking zone of the tested sample.

**Figure 22 materials-16-01316-f022:**
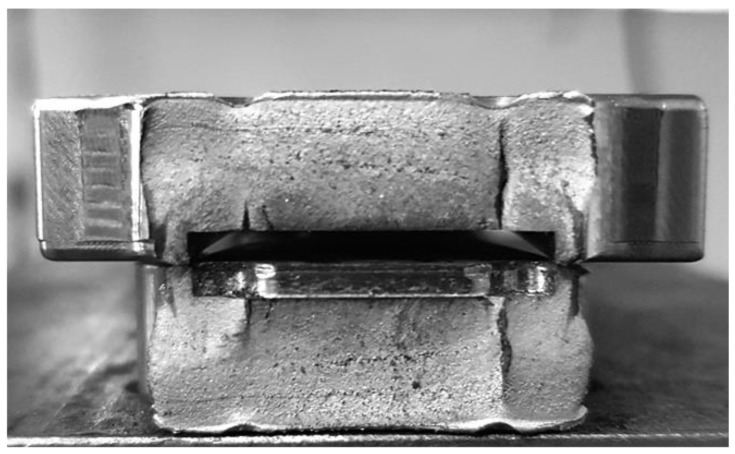
The breaking surfaces of the tested sample.

**Table 1 materials-16-01316-t001:** PZT accelerometers characteristics.

Weight (with Mounting Magnet)	Charge Sensitivity	Voltage Sensitivity	Frequency Range 5%	Max. Cont. Sin Acceleration (Peak)
60 g	4.8 pC/ms^−2^	4 mV/ms^−2^	0.2–5000 Hz	2 · 10^4^ m/s^−2^

## Data Availability

The data of this study are available from the corresponding author upon reasonable request.

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
