# Peer review of "Numerical and Experimental Study of the Fatigue Behavior for a Medical Rehabilitation Exoskeleton Device Using the Resonance Method"

_materials, 2023, doi:10.3390/ma16031316_

Round 1

Reviewer 1 Report

In this paper, the numerical and experimental studies are carried out for the fatigue behavior of a medical rehabilitation exoskeleton device by using the resonance method. It is interesting and within the scope of the journal. Some important issues need to be addressed before the manuscript is accepted.

1.     There are some poor word choices in this paper, and it would be better if the authors could polish the language further. Please revise the format of the images in the text according to the journal's requirements.

2.     In Section “Introduction”, the literature review is not comprehensive. The damage mechanics based approach and machine learning methods are also commonly used for fatigue behavior analysis. See the following papers.

             [1]            Huang J, Meng Q, Zhan Z, et al. Damage mechanics-based approach to studying effects of overload on fatigue life of notched specimens. International Journal of Damage Mechanics, 2019, 28(4): 538-565.

             [2]            Zhan Z, Ao N, Hu Y, et al. Defect‐induced fatigue scattering and assessment of additively manufactured 300M-AerMet100 steel: An investigation based on experiments and machine learning. Engineering Fracture Mechanics, 2022, 264: 108352.

3.     In Section 2.1, it is not clear that why the linear elastic-dynamics problem can be formally equivalent to the variational problem, please give more descriptions.

4.     In Section 2.2, the finite element approximations are presented, and how large is the error induced by the approximation? Figure 2 is not clear enough, please make changes.

5.     In Section 2.3, please ensure that all variables are described. Furthermore, it is not clear how Equation 17 is obtained, please supplement further explanations.

6.     In Section 2.4, “…where the matrix the contains…”Please correct the syntax, and some similar issues throughout the manuscript.

7.     In Section 3.1, Please add a flow chart to better describe the experimental tests. The author mentioned that “a relatively large number of cycles at the resonance frequency (approx. 195Hz)”, It is not clear whether it is reasonable to choose this frequency, especially using for the fatigue cycles.

8.     In Section 3.2, Please supplement more descriptions about the Figs 6~19.

9.     In Section 3.3, in terms of Eq. (19), there are quite a few formulas for calculating damage accumulation, why choose this one? Does it have any advantages over other nonlinear damage accumulation formulas?

10.  In Section 4, please supplement more details about the discussions on the resonance method for testing the fatigue behavior.

11.  In Section 5, suggest writing the summary in points.

Reviewer 2 Report

The article deals with an interesting issue of an interdisciplinary nature. The results of experimental research are valuable and may be of interest to other research teams. The results are interesting both cognitively as well as in terms of application. However, according to the reviewer, the article requires significant changes in terms of its formal structure:
1. Theoretical considerations 2.1 to 2.3 are a restatement of basic knowledge and contribute little to the article. Moreover, the formal notation is unnecessarily complex and inconsistent. I suggest shortening or removing this part from the article as much as possible and referring to the literature.
2. Raw plots of experimental studies are presented. Their number is excessive, and the form requires description and unification. Axes, legends, units, etc. must be described. Limit yourself to a few representative graphs.
3. Standard units should be used, e.g. kg and not Kg (Kelvin times gram???)
4. The results of experimental research should be uniformly summarized in the form of tables and graphs so that interested parties can obtain collective information about the results. The conclusions should also contain a set of relevant, innovative information.
5. The test apparatus should be described in detail, taking into account all relevant parameters, including measurement uncertainty.

Round 2

Reviewer 2 Report

The authors introduced some of the changes and corrections suggested by the reviewer. The article in its current form may be published, however, it is suggested to make linguistic and editorial corrections.